# Risk Factors Associated with the Development of Thrombotic Microangiopathy in Patients with Dermatomyositis

**DOI:** 10.3390/ijms27010315

**Published:** 2025-12-27

**Authors:** Fabiola Cassiano-Quezada, Daniel Alberto Carrillo-Vázquez, Jiram Torres-Ruiz, Nancy Raquel Mejía-Domínguez, Karina Santana-de Anda, Ericka Abigail Guevara-Rojas, Diana Gómez-Martín

**Affiliations:** 1Department of Immunology and Rheumatology, Instituto Nacional de Ciencias Médicas y Nutrición Salvador Zubirán, Mexico City 14080, Mexico; fabiolacassianoq@gmail.com (F.C.-Q.); danielbeatle94@gmail.com (D.A.C.-V.); jiram.torresr@incmnsz.mx (J.T.-R.); karina.santana@incmnsz.mx (K.S.-d.A.); 2Red de Apoyo a La Investigación, Coordinación de Investigación Científica, Universidad Nacional Autónoma de México, Mexico City 14080, Mexico; nmejia@cic.unam.mx; 3Department of Geriatric Medicine, Instituto Nacional de Ciencias Médicas y Nutrición Salvador Zubirán, Mexico City 14080, Mexico; erickagr09@gmail.com

**Keywords:** thrombotic microangiopathy, dermatomyositis, hypocomplementemia, neutrophils, myositis, NETs

## Abstract

Thrombotic microangiopathy (TMA) is an infrequent and poorly understood manifestation in dermatomyositis (DM) associated with poor outcomes and refractoriness to treatment. The aim of this study is to describe the clinical characteristics and risk factors for its development. We conducted a nested case–control study comparing patients with DM who developed TMA to those with DM without this complication. Disease activity was evaluated using the Myositis Disease Activity Assessment Tool (MDAAT), the Manual Muscle Test of eight muscle groups (MMT8), and muscle enzyme levels. A binomial logistic regression analysis was performed to identify risk factors for the development of TMA among patients with DM. All patients with TMA had DM. Patients with DM/TMA had a shorter time since disease onset (*p* = 0.033), lower levels of C3 (*p* = 0.07) and C4 (*p* = 0.046), as well as higher leukocyte (*p* = 0.044), neutrophil (*p* = 0.033), and creatine phosphokinase (CK) levels (p = 0.005). They also exhibited higher constitutional (*p* = 0.0008), pulmonary (*p* = 0.008), and muscle disease activity (*p* = 0.027). In the univariate analysis, a shorter time since disease onset (OR 0.42, *p* = 0.0042) indicated an increased risk for TMA, as did low complement levels (C3: OR 1.11, *p* = 0.01; C4: OR 1.18, *p* = 0.02) and higher constitutional (OR 2.27, *p* = 0.0014), pulmonary (OR 5.50, *p* = 0.0004), and muscle disease activity (OR 2.1, *p* = 0.003). Although elevated CK levels (OR 1.001, *p* = 0.0008) reached statistical significance, the effect size was minimal and should not be interpreted as a clinically relevant increase in risk. Confocal microscopy of muscle biopsy specimens demonstrated neutrophil extracellular traps (NETs) infiltrating muscle tissue. Patients with DM who develop TMA appear to exhibit a distinct clinical phenotype characterized by leukocytosis, neutrophilia, hypocomplementemia, shorter disease duration, and greater constitutional, pulmonary, and muscular disease activity. Although limited by the small sample size, these findings suggest a potential role of NETs in microvascular and tissue injury associated with DM-related TMA. Larger studies are warranted to validate these observations and further elucidate the underlying pathogenic mechanisms.

## 1. Introduction

Dermatomyositis (DM) is a subtype of idiopathic inflammatory myopathy (IIM) characterized by inflammatory skin changes and myositis [1]. The perimysial inflammation, the immune-mediated vascular damage, and the overexpression of type I interferon (IFN) are key pathogenic features of this disease [2,3]. The deleterious effects of the type I IFN manifest as perifascicular atrophy and endothelial damage [4,5], promoting changes in the microcirculation architecture and permeability.

The clinical features associated with vasculopathy are frequent in DM patients and include calcinosis, Raynaud’s phenomenon, livedo reticularis, and leukocytoclastic vasculitis in severe cases [1,6]. In patients with DM, the myositis specific antibodies (MSA) are associated with distinctive clinical features, and some of them show a prominent vasculopathic phenotype. This is the case for anti–TIF1-γ–positive dermatomyositis (DM) patients, who typically present with clinical erythroderma and extensive psoriasiform and hyperkeratotic plaques, as well as for anti–MDA5–positive DM patients, who often develop necrotic cutaneous ulcers, telangiectasias, poikiloderma, and diffuse interstitial lung disease (ILD). Notwithstanding the intense vascular damage in DM, hematologic features are not common among these patients. Particularly, thrombotic microangiopathy (TMA) has only been documented in isolated case reports [7,8,9].

TMA is defined as microangiopathic hemolytic anemia (MAHA), and non-immune thrombocytopenia associated with tissue ischemic damage due to disseminated thrombosis. TMA is a life-threatening complication observed in systemic lupus erythematosus (SLE), antiphospholipid syndrome (APS), and systemic sclerosis (SSc) [10]. Previous studies have assessed the risk factors for TMA in connective tissue diseases. For instance, in SLE patients, the SLE disease activity index (SLEDAI) score, coexisting nephritis, and lymphopenia are the most important independent risk factors for the development of TMA [11,12], but studies assessing the clinical features of patients with DM-associated TMA are lacking. TMA has an incidence of approximately 1.9% in patients with idiopathic inflammatory myopathies (IIM) and is characterized by a heterogenous and atypical clinical presentation, harboring a high risk of treatment refractoriness (37.5%) and death (81.2%) [7]. Therefore, the timely identification of IIM patients at risk of having a TMA is fundamental to immediately initiating appropriate treatment. The aim of this study is to analyze the risk factors for TMA among patients with DM.

## 2. Results

All patients included in this study were Hispanic and had DM. Between 2018 and 2023, patients were evaluated at our center as cases occurred during that period. Most patients were female. Compared with patients without TMA, those with this complication had a shorter median time since disease onset (2.5 months [95% CI, 2–6] vs. 8 months [95% CI, 6–12]; *p* = 0.023). The median age at disease onset was 36.5 years [95% CI, 21.7–50.5] in the TMA group and 52 years [95% CI, 39–67] in the non-TMA group (*p* = 0.145).

Regarding the MSA status, two patients with DM-associated TMA had anti-TIF1-γ, one had anti-Mi2, and one had anti-NXP2 antibodies. In the DM group without TMA, the positivity for MSA was as follows: anti-TIF1-γ, N = 3, anti-Mi2, N = 3, anti NXP2, N = 2, and anti-MDA5, N = 1. We did not find an association between the development of TMA and MSA. As shown in Table 1, patients with DM-associated TMA had a shorter time since disease onset (*p* = 0.003), and higher constitutional (*p* = 0.009), pulmonary (*p* = 0.008) and muscle disease activity (*p* = 0.027). Also, this group presented leukocytosis (*p*= 0.044), neutrophilia (*p* = 0.003), higher CK levels (*p* = 0.005), and had C3 (*p* = 0.07) and C4 (*p* = 0.046) hypocomplementemia in comparison to DM patients without TMA (Table 1). Other specific features of MAHA—defined as a hematologic manifestation resulting from mechanical fragmentation of erythrocytes as they traverse damaged or partially occluded microvessels, characterized by schistocytes, elevated lactate dehydrogenase (LDH), indirect hyperbilirubinemia, low haptoglobin, and reticulocytosis—were not statistically different between groups, mainly due to the limited sample size and incomplete measurements across cohorts. For instance, haptoglobin levels were <30 g/L in all patients with TMA, but this parameter was measured in only three patients in the control group, all of whom had normal values (101–290 g/L). The direct Coombs test was available in three patients with DM-associated TMA; it was negative in two and positive in one, although this latter result was obtained after intravenous immunoglobulin (IVIg) infusion, which may confound interpretation. ADAMTS13 activity was assessed in two patients diagnosed with TMA. The measured activity levels were 41% in the patient positive for anti–TIF1-γ antibodies and 50% in the patient with anti–Mi-2 positivity, indicating only a partial reduction in enzyme activity. Importantly, inhibitory autoantibodies against the von Willebrand factor–cleaving protease (ADAMTS13 inhibitor) were negative in both cases, suggesting that the observed TMA was unlikely related to acquired thrombotic thrombocytopenic purpura (TTP) mediated by severe ADAMTS13 deficiency.

All patients with DM and TMA received pulse methylprednisolone therapy (1 g/day) administered for 3 days in two patients and 5 days in the remaining cases, followed by therapeutic plasma exchange (TPE) in two patients until platelet count recovery was achieved. The anti–Mi-2–positive patient underwent 10 TPE sessions, whereas the anti–TIF1-γ–positive patient required 20 sessions to achieve hematologic improvement. Two patients, one positive for anti–NXP2 and the other for anti–TIF1-γ antibodies, received methotrexate (25 mg/week), with an improvement in platelet count observed in both cases one month after initial pulse methylprednisolone therapy. These patients were evaluated in 2017 and 2019, when methotrexate was selected as the immunosuppressive agent primarily due to limited drug availability, economic constraints, and the more limited understanding of DM and their optimal management at that time. Consequently, treatment decisions reflected both institutional resource considerations and the prevailing therapeutic approaches during that period. In the univariate logistic regression analysis, a longer disease duration, reduced complement levels, elevated CK, and the presence of constitutional, pulmonary, and muscle disease activity were associated with a diagnosis of TMA (Table 2). However, due to the small sample size (N = 13), a multivariate analysis could not be performed. Therefore, these results should be interpreted as exploratory and hypothesis-generating rather than confirmatory. Notably, histopathological evidence of TMA within muscle tissue was identified in patients with DM who also exhibited systemic manifestations of TMA, as illustrated in Figure 1.

## 3. Discussion

To our knowledge, this study is the first to describe the risk factors associated with the development of TMA in the context of active DM. It provides the first systematic characterization of the clinical and biochemical features linked to TMA in DM. The presence of TMA in this setting was associated with a shorter disease duration, greater constitutional, pulmonary, and muscular disease activity, and laboratory evidence of hemolysis, hypocomplementemia, and thrombocytopenia. TMA in DM represents a catastrophic clinical entity, with an estimated survival rate of 18.8% and a treatment refractoriness rate of 37.5% [8]. Previous case reports have linked the development of TMA to the presence of anti–MDA5 antibodies and high disease activity [13]. This antibody is strongly associated with vasculopathy and the expression of a type I interferon (IFN) signature. However, our observations indicate that TMA is not exclusive to anti–MDA5 DM, as none of the patients with TMA in our cohort tested positive for anti–MDA5 antibodies. This finding underscores that TMA may occur across different autoantibody subsets, suggesting a broader pathogenic spectrum within DM. Three of the four patients with DM-associated TMA exhibited a clinical phenotype consistent with the features typically reported for their corresponding myositis-specific antibodies (MSAs). Both anti–TIF1-γ–positive patients presented with extensive rash and dysphagia; however, in contrast to the usually mild muscle involvement described in the literature, they displayed severe myositis. The anti–NXP2–positive patient showed prominent muscle involvement, accompanied by subcutaneous edema and calcinosis [1,13]. The anti–Mi-2–positive DM patient exhibited an atypical clinical phenotype characterized by poor response to treatment, autoimmune encephalitis, and dysphagia. Our findings demonstrate that TMA can occur in DM patients regardless of their myositis-specific antibody (MSA) profile, highlighting instead the importance of independent risk factors such as a shorter disease duration, constitutional, pulmonary, and muscular disease activity, leukocytosis, neutrophilia, elevated CK levels, and hypocomplementemia. These features may help clinicians identify DM subgroups at higher risk for developing TMA.

A shorter time since disease onset and higher MDAAT scores across multiple domains appear to be key determinants of TMA development in DM. The combination of severe constitutional, pulmonary, and muscular activity may prompt patients to seek medical attention earlier, suggesting that a more pronounced systemic proinflammatory phenotype could predispose to DM-associated TMA. Another possible explanation for the increased muscle disease activity and elevated CK levels observed in these patients is ischemic muscle injury secondary to disseminated microthrombosis, as evidenced by histopathologic signs of TMA in muscle biopsy specimens from affected individuals [14].

The proinflammatory anaphylatoxins C3a and C5a can activated innate immune cells such as neutrophils, with a consequent secretion of IL-1B, IL-6 and IL-8, which are constitutive precursors of NETs, amplifying this loop of vascular inflammation enhancing integrins such as CD11b and ICAM-1 [15]. Complement proteins have been related to the pathogenesis of connective tissue-associated TMA, due to the endothelial damage, C5-mediated hypercoagulability and microthrombi formation in small blood vessels, leading to a partial or complete vascular obstruction due to membrane attack complex (MAC)-mediated endothelial injury [14]. In concordance to previous studies, we described hypocomplementemia as a surrogate of disease activity in dermatomyositis [15,16], and in patients with DM-associated TMA. The role of complement activation in the pathogenesis of this complication warrants further investigation. Given the marked complement pathway activation observed in this study, overlap syndromes—particularly systemic lupus erythematosus (SLE)—were excluded based on clinical, serological, and histopathological data according to the 2019 ACR/EULAR classification criteria for SLE [17]. It is important to note that hypocomplementemia is a hallmark of several systemic rheumatic diseases mediated by immune complexes, including but not limited to SLE, Sjögren’s disease, scleroderma, and IgG4-related disease [18]. All of the patients included in this cohort fulfilled the 2017 ACR/EULAR classification criteria for idiopathic inflammatory myopathies, and all had confirmatory muscle biopsy findings [19].

In our study, patients with DM-associated TMA presented an increased total neutrophil count. Neutrophil expansion in IIM is a disease activity biomarker and an adverse prognostic indicator [20]. Notably, an increase in low-density granulocytes, a neutrophil subset associated with enhanced production of neutrophil extracellular traps (NETs), has been linked to various clinical features in active disease, including dermatosis, calcinosis, cutaneous ulcers, vasculopathy, and dysphagia [14]. We demonstrated NETs identified by confocal microscopy in the inflammatory infiltrates of muscle biopsy; this finding provides direct evidence of neutrophil activation and extracellular trap formation within affected muscle tissue, highlighting the contribution of innate immune mechanisms to local tissue damage and sustained inflammation in idiopathic inflammatory myopathies. Therefore, the potential role of neutrophils and NETs on the development of IIM-associated TMA is another exciting hypothesis that requires further investigation.

Our study has many limitations. It is a single center study including a limited number of Hispanic patients. The detection of MSA was performed exclusively by line blot. Although this method is convenient and widely used in clinical practice, its accuracy has been questioned when compared to more specific techniques such as ELISA and immunoprecipitation. It should also be noted that, to date, MSA testing is not routinely performed using ELISA or immunoprecipitation in most laboratories, which further limits the validation of these results. The findings are exploratory and hypothesis generative, because no multivariate analysis could be performed due to sample size constraints Nonetheless, due to the rarity of this manifestation and the long-term somber prognosis of TMA, we consider our data to be helpful to guide clinicians to promptly identify DM patients who would be at risk of developing such a catastrophic entity and to initiate a thorough hematologic follow up and a prompt therapeutic strategy if a high index of suspicion exists.

## 4. Materials and Methods

We conducted a nested case–control study including thirteen patients with DM who fulfilled the 2017 American College of Rheumatology (ACR)/European Alliance of Associations for Rheumatology (EULAR) classification criteria. All participants were part of the Myositis Translational Research Cohort Salvador Zubirán (MYOTReCSZ), with continuous follow-up at the Instituto Nacional de Ciencias Médicas y Nutrición Salvador Zubirán, a tertiary care center in Mexico, between 2017 and 2024.

TMA was diagnosed by the treating physician according to the International Working Group of Clinical Syndromes of Transplant-Associated TMA (European Group for Blood and Marrow Transplantation [EBMT] and European LeukemiaNet criteria), requiring the presence of at least two of the following: negative Coombs test, increased indirect bilirubin, elevated lactate dehydrogenase (LDH), reticulocytosis, decreased haptoglobin, and/or >1% schistocytes in peripheral blood [19,20]. As a disease control group, we included nine patients with dermatomyositis, matched by age, sex, and disease activity (particularly by MMT8 score), without a history of microangiopathy or hemolysis.

Two certified rheumatologists (J.T.R. and D.G.M.) evaluated disease activity using the core set measures of the International Myositis Assessment and Clinical Studies Group (IMACS), including the patient’s and physician’s visual analog scales (VAS) of disease activity, the Manual Muscle Test of eight muscle groups (MMT8), the Myositis Disease Activity Assessment Tool (MDAAT), and the Health Assessment Questionnaire Disability Index (HAQ-DI). The evaluations were not performed in a blinded manner.

The following laboratory parameters were recorded: hemoglobin, hematocrit, mean corpuscular volume, reticulocyte production index, leukocytes, lymphocytes, platelets, globulins, total bilirubin, indirect bilirubin, aspartate aminotransferase (AST), alanine aminotransferase (ALT), international normalized ratio (INR), lactate dehydrogenase (LDH), ferritin, creatine phosphokinase (CK), aldolase, C3, C4, B-type natriuretic peptide (BNP), and troponins. Myositis-specific antibodies (MSA) and myositis-associated antibodies (MAA) were assessed using a commercial line blot immunoassay (EUROIMMUN). Missing data from the unmeasured hemolysis profile such as haptoglobin or Coombs test were not included in the formal statistical analysis. Muscle biopsies were obtained from the left deltoid in all patients.

For confocal microscopy, paraffin-embedded muscle-tissue sections were deparaffinized. After epitope retrieval with the IHC-TEK epitope retrieval solution (IHC WORLD Life Science Products & Services, Ellicott City, MD, USA), slides were blocked for one hour with 10% BSA. Then, the slides were incubated overnight at 4 °C with rabbit anti–human citrullinated histone H3 (Abcam, Cambridge, UK; dilution 1:750), and mouse anti–human LL37 (Santa Cruz Biotechnology, Dallas, TX, USA; 1: 100) diluted in 5% BSA. After washing 3 times with PBS, secondary Abs were added (donkey anti–rabbit Alexa Fluor 488, 1:500, donkey anti–mouse Alexa Fluor 594 [1:500 in 5% BSA], all from Thermo Fisher Scientific, Waltham, MA, USA). Then, the slides were incubated for 10 min at room temperature with 1:1000 Hoechst 33,342 (Thermo Fisher Scientific) and mounted with ProLong Gold Antifade (Thermo Fisher Scientific). The samples were acquired in an Eclipse Ti-E Nikon confocal microscope (Nikon, Minato, Tokyo, Japan).

The study was approved by the institutional research and ethics committees (Ref. 2984). All patients signed a written informed consent before inclusion.


*Statistical Analysis*


Quantitative variables were expressed as medians and interquartile ranges (IQR). We compared the medians using the Mann–Whitney U test. We assessed the association between qualitative variables using the Chi-square test. A logistic regression analysis with calculation of odds ratios (ORs) and 95% confidence intervals (95% CI) was carried out to assess the association between clinical and laboratory features and the development of thrombotic microangiopathy. A *p* value < 0.05 was considered as statistically significant. Statistical analysis was performed with support of the GraphPad Prism version 8.0 (GraphPad Software San Diego, CA, United States of America).

## 5. Conclusions

In summary, given the small sample size, the conclusions of this study should be interpreted with caution. Our findings suggest that patients with DM who develop thrombotic microangiopathy TMA exhibit a distinct clinical phenotype characterized by leukocytosis, neutrophilia, hypocomplementemia, shorter disease duration, and greater constitutional, pulmonary, and muscular activity. These features should alert clinicians to the possibility of TMA in DM patients presenting with anemia and thrombocytopenia.

There is a potential role for NET formation in the inflammatory and microvascular injury underlying TMA in DM. Further multicenter studies are needed to clarify the contribution of neutrophils, NETs, and complement activation to the pathogenesis of DM-associated TMA, and to identify novel biomarkers and potential therapeutic targets.

## Figures and Tables

**Figure 1 ijms-27-00315-f001:**
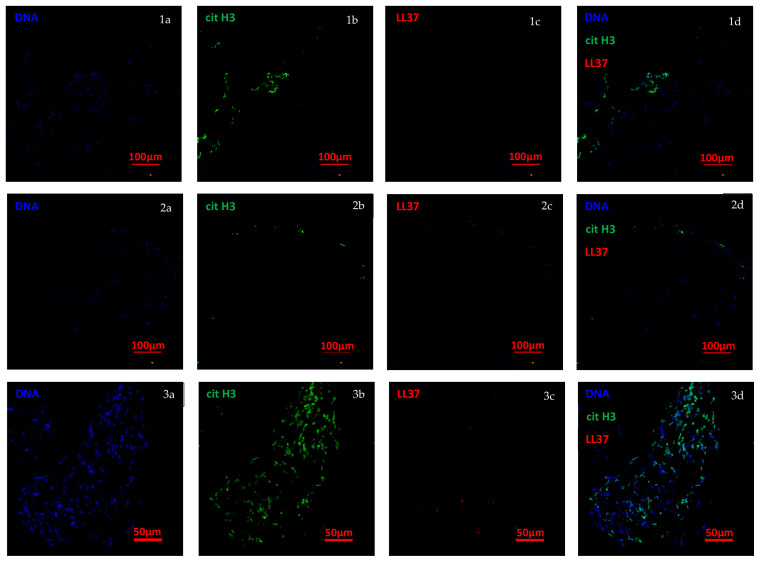
Representative immunofluorescence confocal image of a muscle biopsy showing neutrophil extracellular traps (NETs) infiltrating the muscle tissue of dermatomyositis patient. Blue represents DNA (Column **1a**, **2a**, **3a**), green represents citrullinated histone H3 (**1b**, **2b**, **3b**), and red represents LL37 (**1c**, **2c**, **3c**). Right column images depict merged images for each tissue (**1d**, **2d**, **3d**). Original magnification, ×10. Scale bar 100 µm in the first two rows and 50 µm in the third one.

**Table 1 ijms-27-00315-t001:** Comparison of clinical and laboratory features in patients with DM according to the diagnosis of thrombotic microangiopathy.

Variables	DM and TMA (N = 4) Median (IQR)	DM Without TMA (N = 9) Median (IQR)	*p* Value
Age (years)	36.5 (21.7–50.5)	52 (39–67)	0.145
Gender (Female/Male)	3/1	5/4	0.506
Constitutional activity MDAAT (Score)	12 (7.5–12)	4 (2–5)	*0.009*
Pulmonary activity MDAAT (Score)	10 (8–12)	1 (0–4)	*0.008*
Muscle activity MDAAT (Score)	16 (16–16)	12 (7–15.5)	*0.002*
Cardiovascular MDAAT (Score)	5 (0–12)	1.7 (0–8)	0.157
Gastrointestinal MDAAT (Score)	8 (8–11)	4 (2–9)	0.163
Extramuscular MDAAT (Score)	30 (10–38.25)	11 (7–32.5)	0.345
Time since disease onset (months)	2.5 (2–6)	8 (6–12)	*0.033*
Manual muscle Test 8 (MMT8)	43.5 (15–58)	69.1 (36–115)	0.148
Total leukocytes (×10^3^/µL)	12 (8.1–23.5)	6.4 (4.6–8.5)	*0.044*
Total neutrophil count (×10^3^/µL)	10.98 (6.8–22.2)	4.4 (3.2–6.1)	*0.033*
Total lymphocyte count (×10^3^/µL)	1.35 (0.44–2.27)	0.85 (0.32–1.51)	0.414
Neutrophil to lymphocyte ratio (AU)	19.71 (3.8–47.4)	15.5 (1.7–95.8)	0.335
C3 (mg/dL)	105 (85.5–120.8)	145 (112–148)	*0.07*
C4 (mg/dL)	20 (11–5–30.7)	40 (26–43)	*0.046*
CPK (U/L)	5735 (3952–8299)	1085 (285.5–2340)	*0.005*
Total bilirubin (mg/dL)	1.77 (1.54–1.95)	0.90 (0.57–1.05)	*0.006*
Indirect bilirubin (mg/dL)	1.21 (1.13–1.30)	0.58 (0.38–0.74)	*0.006*
Hemoglobin (g/dL)	8.8 (6.8–11.4)	14.9 (12–8–15.45)	*0.005*
Hematocrit (%)	27.5 (20.28–34.43)	42.3 (37.4–46.8)	*0.016*
Mean Corpuscular Volume (fL)	87.6 (79.3–93.8)	87.1 (84.1–92.53)	0.933
Platelets (×10^3^/µL)	52,000 (42,000–66,750)	217,000 (157,000–287,000)	*0.002*
Ferritin (ng/mL)	1384 (1252–2405)	989 (92.5–4978)	0.4140
Globulins (g/dL)	3.085 (2.32–4.01)	2.94 (2.68–3.27)	0.939
Alanine aminotransferase (U/L)	154 (93.5–306)	84 (44.7–230)	0.330
Aspartate aminotransferase (U/L)	291.5 (213–445.8)	124 (82.95–204.7)	0.075
Lactate dehydrogenase (U/L)	1243 (938–2254)	538 (373–1082)	0.100
Creatinine (mg/dL)	0.72 (0.51–1.275)	0.28 (0.2–0.64)	0.139

DM: dermatomyositis, TMA: Thrombotic microangiopathy, MDAAT: Myositis disease activity assessment tool. The Mann–Whitney U test was used to compare between medians of different populations.

**Table 2 ijms-27-00315-t002:** Clinical and laboratory features associated with the development or protection of TMA in patients with DM.

Variables	Odds Ratio (OR)	95% Confidence Interval (95% CI)	*p* Value
Time since disease onset (months)	0.42	(0.082–0.083)	0.0042
Low C3 (mg/dL)	1.11	(1.02–1.35)	0.01
Low C4 (mg/dL)	1.18	(1.02–1.51)	0.02
Constitutional MDAAT (Score)	2.27	(1.26–10.25)	0.0014
Pulmonary MDAAT (Score)	5.50	(5.17–NA)	0.0004
Muscle MDAAT (Score)	2.1	(7.22–NA)	0.003
CPK (U/L)	1.001	(1.0002–1.0033)	0.0008
Total bilirubin (mg/dL)	4.87	(0–NA)	0.0006
Indirect bilirubin (mg/dL)	2.91	(NA–NA)	0.0006
Hemoglobin (g/dL)	0.23	(0.005–7.029)	0.0008
Platelets (×10^3^/µL)	0.99	(0.48–1.17)	0.0006

## Data Availability

The authors confirm that the data supporting the findings of this study are available for further revision, making a request to the corresponding author.

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
