# Peer review of "Risk Factors Associated with the Development of Thrombotic Microangiopathy in Patients with Dermatomyositis"

_ijms, 2025, doi:10.3390/ijms27010315_

Round 1

Reviewer 1 Report

Comments and Suggestions for Authors
  • ‘ACR’ and ‘EULAR’ should be spelt out on each first mention. This journal is not only for rheumatologists.
  • Line blot is convenient but questioned for its accuracy compared to other methods such as immunoprecipitation and ELISA. Were not the results of MSAs validated by other methods? If not, it would be also noted as limitation.
  • From which parts of body were subjects of muscle biopsy?
  • Information on immunosuppressants administerd for the patients should be provided. Calcineurin inhibitors, which might be a cause of TMA, is especially important so time interval between introduction of cyclosporin or taclorimus and onset of TMA deserves discussed.

Author Response

Reviewer 1

  • ‘ACR’ and ‘EULAR’ should be spelt out on each first mention. This journal is not only for rheumatologists.

Thank you for the suggestion. The full names of the American College of Rheumatology (ACR) and the European Alliance of Associations for Rheumatology (EULAR) have been added upon their first mention in the manuscript.

  • Line blot is convenient but questioned for its accuracy compared to other methods such as immunoprecipitation and ELISA. Were not the results of MSAs validated by other methods? If not, it would be also noted as limitation.

We appreciate this insightful comment. Currently, there is no validated method for detecting myositis-specific antibodies (MSAs) by ELISA or immunoprecipitation. It is well recognized that immunoblot assays may yield false-positive results; therefore, we have added this clarification to the limitations section of the manuscript.

  • From which parts of body were subjects of muscle biopsy?

Thank you for this observation. We have added the information to the Methods section, indicating that all muscle biopsies were obtained from the left deltoid. This site was chosen because it represented a proximal muscle with clinical weakness, and all patients were right-handed.

  • Information on immunosuppressants administered for the patients should be provided. Calcineurin inhibitors, which might be a cause of TMA, is especially important so time interval between introduction of cyclosporin or tacrolimus and onset of TMA deserves discussed.

Thank you for this valuable comment. Among the four patients with dermatomyositis (DM) and thrombotic microangiopathy (TMA), two were not receiving any immunosuppressive therapy at the time of diagnosis, while the other two were being treated with methotrexate. None of the patients had been exposed to calcineurin inhibitors.

Reviewer 2 Report

Comments and Suggestions for Authors

The study was defined by the authors as nested case-control study to compare the differences in clinical as laboratory manifestations between patients with dermatomyositis with and without thrombotic microangiopathy. There are several comments as follow:

  1. All the studied cases who were investigated are dermatomyositis patients. So, the “idiopathic inflammatory myopathies” in the title had better be changed to “dermatomyositis”.
  2. In abstract, the abbreviation “DM” firstly appears in Methods without listing its original name “dermatomyositis” but is only shown original “dermatomyositis” until in Results.
  3. The so-called “constitutional disease activity” was not clearly defined, nor were MDAAT, MITAX, MMT8. Besides, MITAX was not described in the whole course of study. This would lead to nonsense of the comparison of these parameters.
  4. The odds ratio (OR) gives < 1.0 or >1.0 from time to time. However, these were all interpreted as increased risk rather than decreased risk or only been claimed unclearly as “associated”. One more point, the OR of high CK was only 1.001 which cannot be interpreted as increased risk.
  5. The statistical methods used to analyze data in Table 1 should be clearly stated. Moreover, the sample size of TMA (+) was only 4 which may lead to unreliability of statistical analyses.
  6. The figure showing confocal microscopic images are unclear since the columns and rows were not clearly defined (e.g., a,b,c,d, or 1,2,3,4). In addition, the presence of NET formation is almost invisible.
  7. In line 98, the term MAHA abruptly appears without well defining. The term thrombotic microangiopathy and microangiopathic hemolytic anemia should be clearly differentiated.
  8. The ADAMTS13 was only tested in 2 patients. It is inappropriate to claim that 41% with TIF1-gamma and 50% with Mi-2.
  9. The rationale of prescription of MTX in NXP2 and TIF1-gamma patients should be given.
  10. In summary, since the study sample sizes are too small, the conclusions cannot be made so surely. It should be more conservative.
  11. Minor points: tables should not contain vertical lines and should only contain 3 horizontal lines. A too busy table could be split into 2 or 3.
Comments on the Quality of English Language

Some more polish may be needed. 

Author Response

Reviewer 2

The study was defined by the authors as nested case-control study to compare the differences in clinical as laboratory manifestations between patients with dermatomyositis with and without thrombotic microangiopathy. There are several comments as follow:

  • All the studied cases who were investigated are dermatomyositis patients. So, the “idiopathic inflammatory myopathies” in the title had better be changed to “dermatomyositis”.

Thank you for the valuable suggestion. We have modified the title to specify “dermatomyositis” instead of “idiopathic inflammatory myopathies,” as all included cases correspond to patients with dermatomyositis.

  • In abstract, the abbreviation “DM” firstly appears in Methods without listing its original name “dermatomyositis” but is only shown original “dermatomyositis” until in Results.

We appreciate this observation. The full term “dermatomyositis (DM)” has now been introduced upon its first mention in the Abstract, in accordance with journal style guidelines.

  • The so-called “constitutional disease activity” was not clearly defined, nor were MDAAT, MITAX, MMT8. Besides, MITAX was not described in the whole course of study. This would lead to nonsense of the comparison of these parameters.

Thanks for let us know about the redundancy of MITAX, we decided to eliminate this score in order to promote a comprehensive narrative in the manuscript. The constitutional disease activity is the first panel in the Myositis Disease Activity Assessment Tool 2005 version 2, which is the latest update. In this MDAAT definition, clearly specify the features that characterized constitutional activity such as documented fever >38° Celsius, unintentional weight loss >5% and fatigue, malaise or lethargy. We appreciate your observation and decided to included the reference of MDAAT into the bibliography.

  • The odds ratio (OR) gives < 1.0 or >1.0 from time to time. However, these were all interpreted as increased risk rather than decreased risk or only been claimed unclearly as “associated”. One more point, the OR of high CK was only 1.001 which cannot be interpreted as increased risk.

Thank you for your observation. We decided to change the headline of table 2 rephrasing: Clinical and laboratory features associated with the development or protection of TMA in patients with DM. As we specify in the Results section, due to the small sample size (n = 13), a multivariate analysis could not be performed. Therefore, these results should be interpreted as exploratory and hypothesis-generating rather than confirmatory.

  • The statistical methods used to analyze data in Table 1 should be clearly stated. Moreover, the sample size of TMA (+) was only 4 which may lead to unreliability of statistical analyses.

Thank you for pointing this out. We had decided to included the next paragraph in the foot notes on the table 1: The Mann-Whitney U test was used to compare between medians of different populations. We are aware of the scarcity of statistical power due the small numbers of patients presenting with these catastrophic manifestations. That is the reason that we believe is so important the recognized these atypical features, in order to promote a better description of these patients and also raise the awareness in the physicians that treat this patients. 

  • The figure showing confocal microscopic images are unclear since the columns and rows were not clearly defined (e.g., a,b,c,d, or 1,2,3,4). In addition, the presence of NET formation is almost invisible.

Thank you very much for letting us to know about the needed of a better organization system for the rows and columns. We had implemented a combination of letters and numbers in order to identify correctly each of the channels in confocal microscopy. We disagree in the second paragraph in which you mention that the NETs formation is almost invisible. We kindly ask for the current definition of NETosis, in which citrullinated histone H3 in colocalization with DNA is the gold standard to confocal characterizations on NETing neutrophils. However, maybe you are referring to the lack of staining in the LL37 red channel, which is infortune very few. As the continuing body of evidence clarify, different proteins could decorate the chromatin mesh not only LL37, but also MPO or NE. 

Histones, DNA, and Citrullination Promote Neutrophil Extracellular Trap Inflammation by Regulating the Localization and Activation of TLR4. Tsourouktsoglou, Theodora-Dorita et al. Cell Reports, Volume 31, Issue 5, 107602

  • In line 98, the term MAHA abruptly appears without well defining. The term thrombotic microangiopathy and microangiopathic hemolytic anemia should be clearly differentiated.

Thank you for this important observation. We have now defined the term microangiopathic hemolytic anemia (MAHA) upon its first mention and clarified the conceptual distinction between thrombotic microangiopathy —a pathological process—and MAHA—a clinical–laboratory manifestation characterized by hemolytic anemia with schistocytes, elevated LDH, and low haptoglobin. This clarification has been added to the Methods section.

  • The ADAMTS13 was only tested in 2 patients. It is inappropriate to claim that 41% with TIF1-gamma and 50% with Mi-2.

We appreciate the reviewer’s comment. ADAMTS13 activity was assessed in two patients diagnosed with thrombotic microangiopathy (TMA). The measured activity levels were 41% in the patient positive for anti–TIF1-γ antibodies and 50% in the patient with anti–Mi-2 positivity, indicating only a partial reduction in enzyme activity. Importantly, inhibitory autoantibodies against the von Willebrand factor–cleaving protease (ADAMTS13 inhibitor) were negative in both cases, suggesting that the observed TMA was unlikely related to acquired thrombotic thrombocytopenic purpura (TTP) mediated by severe ADAMTS13 deficiency. Accordingly, the text has been revised to accurately describe these findings and to avoid implying quantitative generalization beyond these two cases.

  • The rationale of prescription of MTX in NXP2 and TIF1-gamma patients should be given.

Thank you for this insightful comment. These patients were evaluated in 2017 and 2019, when methotrexate was selected as the immunosuppressive agent primarily due to limited drug availability, economic constraints, and the more limited understanding of dermatomyositis and its optimal management at that time. Consequently, treatment decisions reflected both institutional resource considerations and the prevailing therapeutic approaches during that period.

  • In summary, since the study sample sizes are too small, the conclusions cannot be made so surely. It should be more conservative.

We agree with the reviewer’s assessment. The discussion and conclusion sections have been revised to adopt a more conservative tone, emphasizing that the small sample size limits the strength and generalizability of the findings.

  • Minor points: tables should not contain vertical lines and should only contain 3 horizontal lines. A too busy table could be split into 2 or 3.

Reviewer 3 Report

Comments and Suggestions for Authors

This is a well-designed and clinically relevant study addressing an underexplored topic: thrombotic microangiopathy (TMA) in patients with dermatomyositis (DM). The manuscript is clear, logically structured, and presents novel insights into clinical and biochemical risk factors associated with this rare but severe condition. The integration of clinical data with confocal microscopy findings on neutrophil extracellular traps (NETs) adds significant translational value.

However, before acceptance, several minor revisions are recommended to improve clarity, reproducibility, and overall readability:

1. Methodological clarity – Please provide additional details in the Materials and Methods section regarding:

    • The total period of data collection (start and end years);

    • Whether evaluators of disease activity were blinded to TMA status;

    • How missing data or unmeasured variables (e.g., haptoglobin, Coombs test) were handled in statistical analyses.

2. Statistical limitations – Since the study includes a small sample (n=13), it would be helpful to add a sentence in the Discussion explicitly acknowledging that the findings are exploratory and that no multivariate analysis could be performed due to sample size constraints.

3. Complement and NETs discussion – The discussion could be strengthened by elaborating on the mechanistic relationship between complement activation, NET formation, and endothelial injury in autoimmune myopathies, highlighting how these pathways may converge in the pathogenesis of TMA.

4. Figures and tables –

In Figure 1, please ensure scale bars and magnification details are visible.

In tables, align decimals consistently and simplify variable names (e.g., “Creatinephosphokinase” → “CK”).

5. Terminology consistency – Ensure consistent use of abbreviations such as DM/TMA, MDAAT, MITAX, and NETs throughout the manuscript. Define each term at first mention in the text and tables.

6. English style – The manuscript is well written, but minor linguistic and stylistic improvements would enhance readability. Suggested edits include improving sentence flow, checking article use (“the”), and ensuring consistent tense (past vs. present).

7. References – The reference list is recent and relevant. You might consider adding one or two 2024–2025 reviews focusing on complement-mediated vascular injury and NETs in autoimmune diseases to further strengthen the scientific background.

This work provides valuable new data on the clinical and immunopathological features of TMA in dermatomyositis. The findings are novel, the methodology is sound, and the results are clearly presented. After addressing the minor points above, the manuscript will make a strong contribution to the literature on autoimmune microangiopathies.

Comments on the Quality of English Language

The manuscript is written in generally clear and professional English. The scientific content is well conveyed, and the terminology used is appropriate for the field. However, minor language polishing is recommended to further improve clarity and fluency.

  • Review sentence structure to avoid overly long or complex sentences.

  • Ensure consistent verb tenses (prefer past tense when describing methods and results).

  • Check the use of definite and indefinite articles (“the,” “a,” “an”), as a few are missing in the text.

  • Standardize abbreviations and ensure each is defined at first mention.

  • A brief professional copyediting or proofreading pass would enhance overall readability and style.

Overall, the English quality is good and does not impede comprehension, but light editorial revision is advisable before final publication.

Author Response

Reviewer 3

This is a well-designed and clinically relevant study addressing an underexplored topic: thrombotic microangiopathy (TMA) in patients with dermatomyositis (DM). The manuscript is clear, logically structured, and presents novel insights into clinical and biochemical risk factors associated with this rare but severe condition. The integration of clinical data with confocal microscopy findings on neutrophil extracellular traps (NETs) adds significant translational value.

However, before acceptance, several minor revisions are recommended to improve clarity, reproducibility, and overall readability:

Methodological clarity – Please provide additional details in the Materials and Methods section regarding:

  • The total period of data collection (start and end years);

Thank you for the comment. The study period has been clarified in the Methods section. Data were collected between 2018 and 2023.

  • Whether evaluators of disease activity were blinded to TMA status;

We appreciate this observation. The evaluators of disease activity were not blinded to the presence or absence of thrombotic microangiopathy. This has been acknowledged and added to the limitations section of the manuscript.

  • How missing data or unmeasured variables (e.g., haptoglobin, Coombs test) were handled in statistical analyses.

Certainly, we did not specify about this topic in the Methods section. We updated a paragraph in this regard: Missing data from the unmeasured hemolysis profile such as haptoglobin or Coombs test were not included in the formal statistical analysis

  • Statistical limitations – Since the study includes a small sample (n=13), it would be helpful to add a sentence in the Discussion explicitly acknowledging that the findings are exploratory and that no multivariate analysis could be performed due to sample size constraints.

Thank you very much, we agree that this seminal work is for hypothesis generation, due the small size sample we had decide to include a statement in the end of the Discussion section that integrate this idea.

  • Complement and NETs discussion – The discussion could be strengthened by elaborating on the mechanistic relationship between complement activation, NET formation, and endothelial injury in autoimmune myopathies, highlighting how these pathways may converge in the pathogenesis of TMA.

Thank you for pointing this out. We added some ideas of complement cascade and how these pathways intricated each other in the so-called concept of “immune thrombosis”. The proinflammatory anaphylatoxins C3a and C5a can activated innate immune cells such as neutrophils, with a consequent secretion of IL-1B, IL-6 and IL-8, which are constitutive precursors of NETs, amplifying this loop of vascular inflammation en-hancing integrins such as CD11b and ICAM-1[14]. Complement proteins have been related to the pathogenesis of connective tissue-associated TMA, due to the endothelial damage, C5-mediated hypercoagulability and microthrombi formation in small blood vessels, leading to a partial or complete vascular obstruction due to membrane attack complex (MAC)-mediated endothelial injury [15]. In concordance to previous studies, we described hypocomplementemia, as a surrogate of disease activity in dermatomyo-sitis[14, 16], and in patients with DM-associated TMA. The role of complement activa-tion in the pathogenesis of this complication warrants further investigation.

 Figures and tables –

  • In Figure 1, please ensure scale bars and magnification details are visible.
  • In tables, align decimals consistently and simplify variable names (e.g., “Creatinephosphokinase” → “CK”).

Thank you for making this observations, we hade performed the changes in the Table 1 in order to had a more efficient reading.

  • Terminology consistency – Ensure consistent use of abbreviations such as DM/TMA, MDAAT, MITAX, and NETs throughout the manuscript. Define each term at first mention in the text and tables.

Thank you very much for your kind observation. We had made a minacious review of each abbreviations along the manuscript. We also had decided to retire the MITAX concept, due to there is no clinical relevance as you previously pointed out. 

  • English style – The manuscript is well written, but minor linguistic and stylistic improvements would enhance readability. Suggested edits include improving sentence flow, checking article use (“the”), and ensuring consistent tense (past vs. present).

We appreciate the reviewer’s positive feedback and helpful suggestions. The manuscript has been carefully revised for linguistic and stylistic improvements, including sentence flow, article usage, and consistency of verb tenses throughout the text.

  • References – The reference list is recent and relevant. You might consider adding one or two 2024–2025 reviews focusing on complement-mediated vascular injury and NETs in autoimmune diseases to further strengthen the scientific background.

This work provides valuable new data on the clinical and immunopathological features of TMA in dermatomyositis. The findings are novel, the methodology is sound, and the results are clearly presented. After addressing the minor points above, the manuscript will make a strong contribution to the literature on autoimmune microangiopathies.

Round 2

Reviewer 1 Report

Comments and Suggestions for Authors

(No further comments)

Author Response

Thank you for your kind revision

Reviewer 2 Report

Comments and Suggestions for Authors
  1. Since C3, C4 levels were lower and there is evidence of NETosis in the pathogenesis, the possibility of SLE or SLE overlap should be taken into consideration.
  2. Since TMA or MAHA is most frequently associated with SLE, the cohort of DM with TMA are younger and more likely to be female, the possibility of SLE or SLE overlap should be seriously considered or at least data for anti-dsDNA, acidic ribosomal P, or Smith Ag should be provided to confirm that there really was no association of lupus or lupus associated myopathy.
  3. The authors should at least discuss the overlap or association of SLE or allied diseases in their arguments. 

Author Response

Thank you for allowing us to resubmit our manuscript. We appreciate the valuable comments and suggestions from the editor. We consider we have addressed each comment and are now providing an edited manuscript and a point-by -point reply to the reviewers. Changes in the manuscript were made up throughout the text.

Reviewer 2:

  1. Since C3, C4 levels were lower and there is evidence of NETosis in the pathogenesis, the possibility of SLE or SLE overlap should be taken into consideration.

Thanks for your comment, we agree that hypocomplementemia might suggest SLE overlap, nonetheless all of our patients, fulfilled ACR/EULAR 2017 classification criteria for idiopathic inflammatory myopathies. Besides all included patients had a confirmatory muscle biopsy  None of them indeed fulfilled SLE current classification criteria.

  1. Since TMA or MAHA is most frequently associated with SLE, the cohort of DM with TMA are younger and more likely to be female, the possibility of SLE or SLE overlap should be seriously considered or at least data for anti-dsDNA, acidic ribosomal P, or Smith Ag should be provided to confirm that there really was no association of lupus or lupus associated myopathy.

We appreciate your comment, as we previously cited none of the included patients fulfilled current SLE classification criteria. Relevant autoantibodies related to SLE were negative and are include in the supplementary table.

  1. The authors should at least discuss the overlap or association of SLE or allied diseases in their arguments. 

Thank you for pointing out this issue. This overlap was discarded based on available clinical, serological and histopathological data, as well as the application of classification criteria for this entity. As kindly suggested, we added a discussion in the manuscript:

“Due to the important complement pathways activation signal demonstrated in this study, overlap syndromes (especially systemic lupus erythematosus, [SLE]) were discarded based on the clinical, serological and histopathological data of the current ACR/EULAR 2019 criteria for SLE[17]. It is relevant to address that hypocomplementemia is a hallmark of a diverse subgroup of systemic rheumatic diseases mediated by immunocomplex, including but not limited to SLE, Sjogren’s disease, scleroderma, IgG4-related disease[18]. All of our patients, fulfilled ACR/EULAR 2017 classification criteria for idiopathic inflammatory myopathies, besides all included patients had a confirmatory muscle biopsy[19]”.

Round 3

Reviewer 2 Report

Comments and Suggestions for Authors
  1. All of the rebuttal paragraph should be included into manuscript per se. Hiding the data would lead readers to be skeptical for the results and the resulting discussion. The rebuttals are not just for reviewers. These arguments should be visible in Discussion for public judgement.
  2. There are still vertical lines in tables.
  3. Statistical symbol should be italic, e.g., p value, p should be italic.
Comments on the Quality of English Language

OK, I have no more comments.

Author Response

1. All of the rebuttal paragraph should be included into manuscript per se. Hiding the data would lead readers to be skeptical for the results and the resulting discussion. The rebuttals are not just for reviewers. These arguments should be visible in Discussion for public judgement.

Thank you for this important comment. We agree that hypocomplementemia could strongly associate our findings with SLE. As suggested, we have incorporated the relevant rebuttal content directly into the Discussion section. The following paragraph has now been added to address this point:

“Given the marked complement pathway activation observed in this study, overlap syndromes—particularly systemic lupus erythematosus (SLE)—were excluded based on clinical, serological, and histopathological data according to the 2019 ACR/EULAR classification criteria for SLE.[17] It is important to note that hypocomplementemia is a hallmark of several systemic rheumatic diseases mediated by immune complexes, including but not limited to SLE, Sjögren’s disease, scleroderma, and IgG4-related disease.[18] All of the patients included in this cohort fulfilled the 2017 ACR/EULAR classification criteria for idiopathic inflammatory myopathies, and all had confirmatory muscle biopsy findings.[19]”

2. “There are still vertical lines in tables.”

Thank you for your comment. We have revised the formatting of all tables and removed the vertical lines as requested.

3. “Statistical symbol should be italic…”

Thank you for noting this. We have carefully revised the manuscript and changed all instances of the statistical symbol p to italic formatting.